# Lateral Inhibition-inspired structure for Convolutional Neural Network on Image Classification

## Abstract

Convolutional neural networks (CNNs) have become powerful and popular tools since deep learning emerged for image classification in the computer vision field. For better recognition, the dimension of both depth and width has been explored, leading to convolutional neural networks with more layers and channels. In addition to these factors, neurobiology suggests lateral inhibition (lateral antagonism, e.g. Mach band effect), a widely existing phenomenon for vision that increases the contrast and sharpness of nearby neuron excitation in the lateral direction to help recognition. However, such mechanism has not been well explored in the design of convolutional neural network. In this paper, we explicitly explore the filter dimension in the lateral direction and propose our lateral inhibition-inspired (LI) structure. Our naive design uses the low-pass filter to mimic the strength decay of lateral interaction from neighbors regarding the distance. One learnable parameter per channel is applied to set the amplitude of the low-pass filter by multiplication, which is flexible to model various lateral interactions (including lateral inhibition). The convolution result is then subtracted from the input, which could increase the contrast and sharpness for better recognition. Furthermore, a learnable scaling factor and shift are applied to adjust the value after subtraction. Our lateral inhibition-inspired (LI) structure works on both plain convolution and the convolutional block with residual connection, while being compatible with the existing modules. Preliminary results demonstrate obvious improvements on the ImageNet dataset for AlexNet (7.58%) and ResNet-18 (0.81%), respectively, with little increase in parameters, indicating the effectiveness of our brain-similar design to help feature learning for image classification from a different perspective.

## 1 Introduction

In recent years, convolutional neural networks (CNNs) (Hinton et al., 2012; Simonyan & Zisserman, 2015; Szegedy et al., 2015; He et al., 2016) have become powerful and popular tools since deep learning emerged for image classification in the computer vision field. They have recorded record-breaking performance and outperformed traditional methods (Quinlan, 1986; Cortes & Vapnik, 1995) with hand-crafted features (Lowe, 1999; Dalal & Triggs, 2005) on the ImageNet Large Scale Visual Recognition Challenge (ILSVRC) (Deng et al., 2009). Today, convolutional neural networks still possess unique merits, as they have been studied the most, along with the fact that convolution has a strong connection with the human vision system and image processing, making them good models for feature learning research.

Different factors have been explored to improve recognition performance of convolutional neural networks. VGGNet (Simonyan & Zisserman, 2015) applies a small convolution kernel size ($3 \times 3$) for increased network depth, while ResNet (He et al., 2016) introduces deep residual learning to make training very deep deep networks feasible. The success of such networks indicates that depth is a crucial factor for recognition performance. Wide Residual Networks (Zagoruyko & Komodakis, 2016), on the other hand, demonstrate width as another important factor to improved performance. In addition to these factors, neurobiology suggests the widely existing lateral inhibition (lateral antagonism, e.g. Mach band effect, shown in Fig. 1), a phenomenon that increases the contrast and sharpness of nearby neuron excitation in the lateral direction, also important to help feature learning.

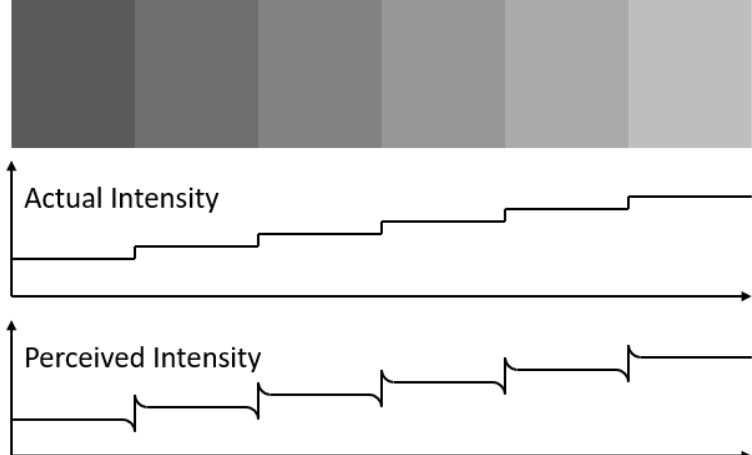

Figure 1: Illustration of Mach band effect. Due to lateral inhibition, the actual perception is different from the real input, leading to increased contrast and sharpness along the boundary (see curves towards opposite direction), where the dark area becomes darker and the bright area becomes brighter.

However, such an important mechanism has not been well explored in convolutional neural network design. In recent years, Yang et al. (2015) propose a traditional method using manual double-opponent design (antagonism) with spatial sparseness constraint for boundary detection, following the color-opponent mechanism in the human vision system, to highlight the color boundary while suppressing texture edges. This double-opponent design is further extended (Gao et al., 2015) for color constancy. Cao et al. (2018) apply lateral inhibition mechanism in the inference stage of the pre-trained VGGNet (Simonyan & Zisserman, 2015) for visual attention map and saliency detection. Hasani et al. (2019) propose their surround modulation design, using a manually defined kernel from the difference of Gaussian (DoG) function, to explore the feasibility of incorporating the lateral inhibition mechanism. This fixed design is only applied to half channels of the initial layer (with better performance than applying to the full channels). It is still worth exploring flexible lateral inhibition-inspired design for convolutional neural network, to incorporate such mechanism and make it more similar to the brain in the training stage for image classification.

In this paper, inspired by the findings of neurobiology, we explicitly explore the filter dimension in the lateral direction and propose our lateral inhibition-inspired (LI) structure, which incorporates the lateral inhibition mechanism in a flexible manner inside modern convolutional neural networks. Our simple naive design applies the low-pass filter with the central weight eliminated, to mimic the decay of inhibition strength from neighbors regarding the distance. To set the amplitude of the low-pass filter by multiplication, a learnable parameter per channel is used, for the flexibility of modeling various lateral interactions (including lateral inhibition). The convolution result is then subtracted from the input, which could increase the contrast and sharpness for better recognition. Furthermore, a learnable scaling factor and shift are applied to adjust the value after subtraction. Our lateral inhibition-inspired (LI) method works on both plain convolution and convolutional block with a residual connection, while being compatible with the existing modules. Preliminary results of our simple naive design demonstrate obvious improvements on the ImageNet dataset for AlexNet (7.58%), ResNet-18 (0.81%), respectively, with little parameter increase, indicating the effectiveness of our brain-similar to help feature learning for image classification from a different perspective.

Our main contributions can be summarized as:

- We propose a lateral inhibition-inspired structure for modern convolutional neural network design, which could participate in the training stage for image classification.
- Our design is flexible compared to the fixed difference of Gaussian (DoG) filter, and could be applied to all layers (and channels) with competitive performance.
- We are the few to explicitly model lateral inhibition as well as other lateral interaction to make the network more brain-similar, given the flexible weight design.

## 2 RELATED WORK

**Neurobiology.** Lateral inhibition is common in sensory system. Mach reports the Mach bands effect RATLIFF (1965) due to lateral inhibition that the human vision system perceives increased contrast and sharpness along the boundary, where the dark area becomes darker and the bright area becomes brighter. Blakemore et al. (1970) discover that lateral inhibition plays an important role in orientation detection within the human vision system. It helps to remove redundancy for the existence of inhibitory signals, as large stimulus covering a whole receptive field would be relatively ineffective in excitation of neurons.

**Traditional method.** Yang et al. (2015) propose their double-opponent design with spatial sparseness constraint for boundary detection, by mimicking the color-opponent mechanism in the human vision system. The single opponent uses a manual antagonism such as red-green and blue-yellow, while the double opponent combines two opposite single opponent in either center-surround or side-by-side manner for increased contrast. This design highlights the color boundary while suppressing texture edges. The double-opponent design is further extended (Gao et al., 2015) for color constancy.

**Convolutional Neural Network.** Local Response Normalization (LRN) (Hinton et al., 2012) is introduced in AlexNetalong and is not trainable. Normalization is performed for the value at each position (x, y) considering neighbors at the same position from nearby channels, with the denominator as a sum of the squared neighbors, given a scaling factor, bias, as well as the exponential value for the denominator. It applies lateral inhibition across nearby channels via competition. Batch normalization (Ioffe & Szegedy, 2015) applies normalization within individual channels, using an mini-batch mean (shift) and variance, then a scaling factor and re-shift parameter, to constrain all values in a certain range. Layer normalization (Ba et al., 2016) only consider one single instance, but performs normalization across all channels using the average and variance from each channel. Group normalization (Wu & He, 2018) further extends layer normalization by dividing channels into groups for consideration. Besides normalization methods, Cao et al. (2018) propose their lateral inhibition design using pretrained VGGNet, as top-down step for attention map and weakly supervised saliency detection. The inhibition value consists of an average term from the neighbors within the same channel to protect high response zone, and a differential term only consider the difference from higher activation nearby for inhibition, with weight decay regarding the distance. The whole formula including term coefficients are intuitively decided and unlearnable, and only happens in inference stage. Hasani et al. (2019) propose surround modulation for lateral inhibition, to make network closer to the brain. It applies two gaussian filters (5x5) with 1:1 weight (fixed) from the difference of Gaussian (DoG) function. However, this fixed design lacks flexibility, which is only applied to half channels of the initial layer within a small convolutional neural network, due to the performance drop when applying to full channels or the following layer. The channelwise normalization (center pool) in Weighted normalization (Pan et al., 2021) improves LRN (Hinton et al., 2012) from AlexNet by adding extra parameters such as a scaling factor and bias, while further adding surround pool for normalization within each channel does not help on ImageNet dataset. DivisiveNorm (Miller et al., 2021) also similarly follows the design of LRN at each location along the channel dimension, but changes the numerator to be squared, and adds a weight scaling factor for each unnormalized activations inside the exponentially weighted sum. The hyper parameters are manually set for stable training.

## 3 METHOD

Our lateral inhibition-inspired (LI) structure explicitly incorporates the lateral inhibition mechanism into the design of convolutional neural network, to increase contrast and sharpness for more accurate perception. Unlike methods only involving in the post-processing stage with the original pre-trained model unchanged, our method participates in the training stage.

Our lateral inhibition-inspired (LI) design takes two steps: first, it generates the lateral interaction value based on the input within the spatial neighborhood of each channel; then it subtracts that value from the original input. Details will be explained in the following parts.

### 3.1 LATERAL INTERACTION DESIGN

We design our naive lateral interaction based on neurobiological findings. It follows the principle that the strength of interaction (including inhibition) from an active neuron decreases as the distance to that neuron increases in the lateral direction. For implementation, we take the exsiting Gaussian low-pass filter with central weight eliminated to mimic the strength decay from neighbors. Here, we denote the function of lateral interaction as *LI*, Gaussian low-pass filter as *G*, the central weight value of the low-pass filter *G* as *C*, convolution operation as *Conv*, the learnable amplitude weights as *W* (with one parameter per channel for multiplication to separately decide the channelwise amplitude of Gaussian low-pass filter *G*), and the input features as *x* for our lateral inhibition-inspired structure.

The weights of the 2D Gaussian low-pass filter (*G*) are calculated using Equation 1. Here $u$ and $v$ denote the spatial coordinates regarding to the center of the Gaussian low-pass filter, and $\sigma$ denotes the standard deviation for weight decay speed from the center (we set filter size as 3 and $\sigma$ as 1):

$$G(u, v) = \frac{1}{2\pi\sigma^2} e^{-\frac{u^2 + v^2}{2\sigma^2}} \tag{1}$$

The lateral interaction function given the input feature *x* at channel *i* is defined in Equation 2. First, convolution on input feature using Gaussian low-pass filter *G* is applied, then the corresponding convolution result form the central weight value C (of the low-pass filter *G*) is subtracted, to only have interaction from the neighbors. Finally, multiplication is performed by the amplitude parameter (from the learnable weights *W*) to decide the amplitude of the Gaussian low-pass filter *G* per channel. Different from Surround Modulation (Hasani et al., 2019), which applies 1:1 weight using two Gaussian low-pass filters in the difference of Gaussian (DoG) function to model lateral inhibition, our learnable weights *W* is more flexible (could be zero, positive or negative) to fit various lateral interactions (for none, lateral inhibition or even the opposite when necessary).

$$LI(x_i) = (Conv(G, x_i) - C * x_i) * W_i \tag{2}$$

### 3.2 APPLYING LATERAL INTERACTION

We denote the output function as *O* for applying lateral interaction on input feature *x* at channel *i*. Then the output function is defined in Equation 3, with lateral interaction result subtracted.

$$O(x_i) = x_i - LI(x_i) \tag{3}$$

### 3.3 ADDITIONAL PARAMETERS

Additional parameters are applied since the interaction result will be subtracted from the original feature map. Parameters include a scaling factor ($v$), shift ($m$), and bias ($b$). Combination of these parameters can be explored to find the best setting.

$$O(x_i) = (O(x_i) + m) * v + b \tag{4}$$

### 3.4 STRUCTURE PLACE

Our lateral inhibition-inspired (LI) structure works on both the plain convolution and the convolutional block with residual connection. For the plain convolution, it is placed after that convolution (and the activation function), as shown in Fig. 2. For the convolutional block with a residual connection, it is placed inside the residual block before the first convolution, as shown in Fig. 2.

## 4 EXPERIMENTS

**Dataset.** We use the ImageNet-1k data set (Russakovsky et al., 2015) to evaluate our proposed methods. ImageNet-1k dataset consists of 1k object categories, with 1.2 million images for training and 50k images for validation.

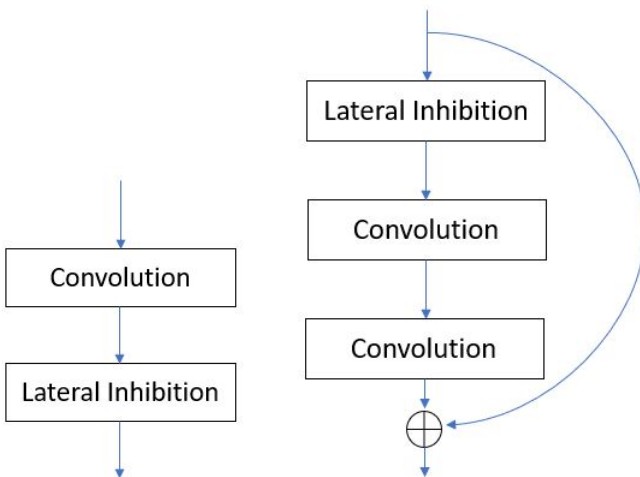

Figure 2: Place for our lateral inhibition-inspired (LI) structure. It is after the plain convolution or inside the residual block before the first convolution, as shown on the left or right side, respectively.

**Experiment Settings.** We use the Pytorch (Paszke et al., 2019) deep learning framework and follow the official training example. We adopt the standard settings: image crop size of $224 \times 224$, initial learning rate of 0.01 for AlexNet (Hinton et al., 2012) (which decays every 30 epochs by a factor of 10), momentum of 0.9, and weight decay of 0.0001. We use standard data augmentation, which includes random horizontal flipping, random scale and ratio changes in the range of [0.08, 1.0] and [0.75, 1.33], respectively. Models are trained with batch size of 128 and batch accumulation of 2. Experiments are performed on 4 GPUs, with 90 epochs for AlexNet following Miller et al. (2021) (more epochs show little difference) and 100 epochs for ResNet following common practice.

**Results.** Table 1 shows the results of the baseline model AlexNet (Hinton et al., 2012), related methods, as well as our naive lateral inhibition-inspired (LI) design on ImageNet dataset. While all listed methods improve the baseline significantly, our method outperforms others by a large margin. Layer normalization only consider one instance, which fits better for dynamic input rather than batch input. Group normalization has further division into groups, therefore limits its performance. Weighted normalization and Divisive normalization are similar in the normalization design, except the power number in the numerator and the denominator, as well as the weighted sum. However, most of them lack the consideration of neighbors within the same channel for lateral interaction, which our method does.

Table 2 shows the results of the baseline model ResNet (He et al., 2016) and our naive lateral inhibition inspired (LI) design on ImageNet dataset. Although ResNet contains batch normalization to improve performance, our design makes the improvement further by 0.81%.

**Design Choice.** From Table 3 we can see that, with a single scaling factor, our design already outperforms other methods. Among all the design variants, scaling first with bias is better than shift first then scaling. While they are both much better than only scaling, the difference between the best two are not large.

## 5  CONCLUSION

In this paper, we propose our lateral inhibition-inspired structure for convolutional neural network to make it more brain-similar on image classification task. We explicitly explore neurobiology findings, using the guassian low-pass filter with central weight eliminated to mimic the strength decay regarding the distance. With a learnable weight, our deign is flexible to model lateral inhibition as well as other lateral interaction. Our design works on both plain convolution and the convolutional block with residual connection, while being compatible with the existing modules. Preliminary results using AlexNet and ResNet demonstrate obvious improvements on the ImageNet dataset for

Table 1: Top-1 accuracy of AlexNet models (with only plain convolution) on ImageNet dataset. LI denotes our lateral inhibition-inspired (LI) design, with little increase in parameters and GFLOPs. Surround Modulation is added after every convolutional layer as other methods in the experiments.

| Method | Top-1 Acc (%) | Params. | GFLOPs | Improv. (abs) |
|---|---|---|---|---|
| AlexNet (baseline) | 52.82 | 62.38M | 1.137 | - |
| +GroupNorm (Wu & He, 2018) | 57.76 | 62.38M | 1.137 | 4.94 |
| +LayerNorm (Ba et al., 2016) | 58.81 | 63.68M | 1.137 | 5.99 |
| +BatchNorm (Ioffe & Szegedy, 2015) | 59.04 | 62.38M | 1.138 | 6.22 |
| +WeightedNorm-c (Pan et al., 2021) | 59.41 | 62.38M | 1.137 | 6.59 |
| +DivisiveNorm (Miller et al., 2021) | 59.54 | 62.38M | 1.137 | 6.72 |
| +LI (ours) | **60.39** | 62.38M | 1.143 | **7.58** |

Table 2: Top-1 accuracy of ResNet18 models (contains residual block) on ImageNet dataset. LI denotes our lateral inhibition-inspired (LI) design, with little increase in parameters and GFLOPs. Our structure is applied to both the initial plain convolution and the following residual blocks. Please note that ResNet already contains batch normalization to increase performance.

| Method | Top-1 Acc (%) | Params. | GFLOPs | Improv. (abs) |
|---|---|---|---|---|
| ResNet18 (baseline) | 70.30 | 11.69M | 1.822 | - |
| +LI (ours) | **71.11** | 11.69M | 1.838 | **0.81** |

image classification, with little increase in parameters. We hope that our research could inspire researchers to reconsider the value of this track for better feature learning.

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

Table 3: Top-1 accuracy of AlexNet models (with only plain convolution) on ImageNet dataset. LI denotes our lateral inhibition-inspired (LI) design. Shift, scaling, and re-shift means that our design variants are changed by adding additional parameters.

| Method | Top-1 Acc (%) | Params. | GFLOPs | Improv. (abs) |
|---|---|---|---|---|
| AlexNet (baseline) | 52.82 | 62.38M | 1.137 | - |
| +LI (scale) | 59.87 | 62.38M | 1.143 | 7.05 |
| +LI (shift-scaling) | 60.26 | 62.38M | 1.143 | 7.44 |
| +LI (scale-bias) | **60.39** | 62.38M | 1.143 | **7.58** |

Hosein Hasani, Mahdieh Soleymani, and Hamid Aghajan. Surround modulation: A bio-inspired connectivity structure for convolutional neural networks. *Advances in neural information processing systems*, 32, 2019.

Kaiming He, Xiangyu Zhang, Shaoqing Ren, and Jian Sun. Deep residual learning for image recognition. In *Proceedings of the IEEE conference on computer vision and pattern recognition*, pp. 770–778, 2016.

Geoffrey E Hinton, Alex Krizhevsky, and Ilya Sutskever. Imagenet classification with deep convolutional neural networks. *Advances in neural information processing systems*, 25:1106–1114, 2012.

Sergey Ioffe and Christian Szegedy. Batch normalization: Accelerating deep network training by reducing internal covariate shift. In *Proceedings of Machine Learning Research*, volume 37, pp. 448–456, 2015.

David G Lowe. Object recognition from local scale-invariant features. In *Proceedings of the seventh IEEE international conference on computer vision*, volume 2, pp. 1150–1157. Ieee, 1999.

Michelle Miller, SueYeon Chung, and Kenneth D Miller. Divisive Feature Normalization Improves Image Recognition Performance in AlexNet. In *International Conference on Learning Representations*, 2021.

X Pan, E Kartal, L G S Giraldo, and O Schwartz. Brain-inspired weighted normalization for CNN image classification. In *International Conference on Learning Representations (ICLR)*, 2021.

Adam Paszke, Sam Gross, Francisco Massa, Adam Lerer, James Bradbury, Gregory Chanan, Trevor Killeen, Zeming Lin, Natalia Gimelshein, Luca Antiga, and Others. PyTorch: An Imperative Style, High-Performance Deep Learning Library. *Advances in Neural Information Processing Systems*, 32:8026–8037, 2019.

J Ross Quinlan. Induction of decision trees. *Machine learning*, 1(1):81–106, 1986.

F RATLIFF. Mach bands: Quantitative studies on neural networks in the retina. *Holden-Day (Holden-Day series in psychology)*, 1965.

Olga Russakovsky, Jia Deng, Hao Su, Jonathan Krause, Sanjeev Satheesh, Sean Ma, Zhiheng Huang, Andrej Karpathy, Aditya Khosla, Michael Bernstein, and Others. Imagenet large scale visual recognition challenge. *International Journal of Computer Vision*, 115(3):211–252, 2015.

Karen Simonyan and Andrew Zisserman. Very deep convolutional networks for large-scale image recognition. In *Proceedings of 3rd International Conference on Learning Representations*, 2015.

Christian Szegedy, Wei Liu, Yangqing Jia, Pierre Sermanet, Scott Reed, Dragomir Anguelov, Dumitru Erhan, Vincent Vanhoucke, and Andrew Rabinovich. Going deeper with convolutions. In *Proceedings of the IEEE conference on computer vision and pattern recognition*, pp. 1–9, 2015.

Yuxin Wu and Kaiming He. Group normalization. In *Proceedings of the European conference on computer vision (ECCV)*, pp. 3–19, 2018.

Kai-Fu Yang, Shao-Bing Gao, Ce-Feng Guo, Chao-Yi Li, and Yong-Jie Li. Boundary detection using double-opponency and spatial sparseness constraint. *IEEE Transactions on Image Processing*, 24(8):2565–2578, 2015.

Sergey Zagoruyko and Nikos Komodakis. Wide residual networks. In *Proceedings of the British Machine Vision Conference*, 2016.

