# OpenReview forum: "Suppression helps: Lateral Inhibition-inspired Convolutional Neural Network for Image Classification"
_ICLR.cc/2023/Conference — Submitted to ICLR 2023_

### Official Review · Reviewer_KkWZ · 2022-10-22

**Confidence:** 5
**Correctness:** 2
**Technical Novelty And Significance:** 1
**Empirical Novelty And Significance:** Not applicable
**Recommendation:** 1

**Clarity, Quality, Novelty And Reproducibility:**

The paper is clear to read. But its quality is below the conference standard. The method is simple and the novelty is limited. The reproducibility seems good.

**Strength And Weaknesses:**

This paper is motivated by a neurobiological effect named lateral inhibition. It is always fascinating to see methods that are inspired in this way because it may reveal some important angles that the academic community has overlooked.

But this paper seems to be less prepared for this conference. The method is interesting but very simple: it adds a couple of convolutional layers to existing architectures. The baseline architectures are not from recent works, and the model scales in the test are also small (e.g. ResNet18, AlexNet). I also didn’t find the visualizations mentioned in the paper. I hope that the authors could add more experiments to the paper, show larger experiment settings, and polish the paper a bit more.

**Summary Of The Paper:**

This paper proposes a type of neural networks that uses lateral inhibition inspired layers. Lateral inhibition in this paper refers to the effect where the contrast of nearby neuron excitation in the lateral direction is increased for recognition. Inspired by this, the paper proposes to use a Gaussian low-pass filter with a zero center weight along with a learnable channel weight to mimic this effect. A flexible alternative would be a depthwise convolution. Then this output is used to subtract from the input tensor to form the final output. This lateral inhibition inspired design is tested with AlexNet and ResNet18 for image classification on ImageNet, and shows improvements at little additional cost.

**Summary Of The Review:**

This is a paper inspired by the lateral inhibition effect. The motivation is great, but the method is too simple, and the paper does not show the strength of the method through large-scale and comprehensive experiments. Overall, I think the paper is below the conference standard.

---

> ### Author Response · Authors · 2022-11-16
> **Comparison with related methods**
>
> Thank you for your time and valuable comments. Idea comes shortly before the initial submission, and we try to present the preliminary results for other researchers. Our paper is about lateral inhibition inspired structure. We have performed comparison with several normalization methods, according to some reviewers' opinion. We have obtained the code from authors of [1] (ICLR 2022), and outperform all of them by a large margin. We will address your concerns accordingly.
>
>  - **The method is interesting but very simple: it adds a couple of convolutional layers to existing architectures.**
>
>     Yes, however, using a filter to be more brain-similar (only need to consider our naive design with guassian filter, it has learnable weight W for the guassian filter, to compare with related methods) from lateral inhibition perspective within the same channel to increase the contrast and sharpness along the boundary (while making homogeneous area less prominent, consider Mach band effect) is rarely explored. We have a learnable weight W for our guassian filter, which is flexible (could be zero, positive or negative) to fit various lateral interactions. *The convolution result will be subtracted from the original feature map.*
>
>
> - **The baseline architectures are not from recent works, and the model scales in the test are also small (e.g. ResNet18, AlexNet).**
>
>   Given limited time and resources, feature learning research may be only verified using small network on ImageNet dataset for image classification, as some brain-inspired methods [1][2] (ICLR 2022, 2021) using only AlexNet. We will do the best we can.
>
> - **I hope that the authors could add more experiments to the paper, show larger experiment settings, and polish the paper a bit more.**
>
>   Yes, we will. We have requested and obtained the code from [2] (and fix their missing padding for the initial convolution) for comparison with full size AlexNet (initially our method uses Pytorch build-in official model for AlexNet, which comes from [3] with less channels). AlexNet is the only model used in [1][2] (ICLR 2022, 2021) on ImageNet dataset. With a simple scaling factor (to be learned) per channel, our method already outperforms all of them; with shift mean or bias (to be learned) per channel makes further improvement. Results on ImageNet dataset (best model for 90 epochs as [1], while 100 epochs show no much difference) for Top-1 Acc with 4 GPU setting:
>
>     - AlexNet (baseline) 52.82%
>     - GroupNorm    57.76%
>     - LayerNorm    58.81%
>     - BatchNorm    59.04%
>     - WeightedNorm-c (center)  59.41%
>     - DivisiveNorm  59.54%
>     - Ours (scale) 59.87%
>     - Ours (shift-scale) 60.26%
>     - Ours (scale-bias) **60.39%**
>
> [1] Miller, M., Chung, S., & Miller, K. D. (2021, September). Divisive Feature Normalization Improves Image Recognition Performance in AlexNet. In International Conference on Learning Representations.
>
> [2] Pan, X., Giraldo, L. G. S., Kartal, E., & Schwartz, O. (2021). Brain-inspired weighted normalization for CNN image classification. In International Conference on Learning Representations.
>
> [3] One weird trick for parallelizing convolutional neural networks. Alex Krizhevsky, 2014.

---

> > ### Comment · Reviewer_KkWZ · 2022-11-29
> > **Acknowledging author responses**
> >
> > Dear Authors,
> >
> > Thank you so much for preparing the response to our initial comments in such a short time! I think we are on the same page about the potential of this method, and the work needed to be done to improve the overall quality of the paper to meet the conference standard. Regarding the results, the lack of large-scale experiments is still concerning given that the proposed method adds complexity to networks. I encourage the authors to further improve the depth and the value of the method, and polish the paper quality to fully release the potential of the idea. I will not recommend acceptance at the current stage but would love to see an improved revision coming out. Thanks again!
> >
> > Cheers

---

### Official Review · Reviewer_GVUz · 2022-10-22

**Confidence:** 5
**Correctness:** 2
**Technical Novelty And Significance:** 1
**Empirical Novelty And Significance:** 1
**Recommendation:** 3

**Clarity, Quality, Novelty And Reproducibility:**


The quality, clarity, and the originality are poor.



**Details Of Ethics Concerns:**


There is no ethics concern.


**Strength And Weaknesses:**


(Positive) Although this article is very poor, its exploratory spirit, specifically the exploratory spirit of exploring neural networks that conform to human cognition, is worthy of recognition.


(Negative) The authors claim some gain in their approach. But as we all know, after introducing attention or dynamic mechanism, a general neural network can have some improvement. In particular, the worse the network, the more obvious the improvement.

(Negative) In fact, filtering is not a new thing in neural networks.


(Negative) A very working method should be validated on large-scale tasks. In addition to classification, there is detection, segmentation, and so on.


(Negative) The writing of this article is very poor. There are multiple copies of the text in the article. For example, there are multiple repetitions of text in the abstract and introduction. A good article should have a better way of expressing it.


(Negative) Gaussian filtering and bilateral filtering are not new in neural networks. The method proposed in the article is equivalent to a bilateral filtering non-local neural network.

(Negative) The article associates its method with depth-wise convolution. This is very imprecise and arrogant. If this is true, is MLP-Mixer also a special case of this article?

(Negative) As I said before, adding attention or dynamics to the neural network, such as senet, sknet, acnet, dynamic nets, will bring gains. There are too many such examples. So the results in Table 2 are not dazzling at all.

(Negative) The numbers in Equation 5 are so intuitive and natural.



**Summary Of The Paper:**


Summary:
This article starts from the perspective of neuroscience, hoping to design a neural network that conforms to human cognition. The original intention of the article is good, but there is nothing special about the method of the article. Regarding the writing, methodology, and experiments, this article is far below the acceptance criteria for a good conference like ICLR. Reviewers put a lot of effort into reviewing each article, so hopefully, authors will write each article well before submitting it.



**Summary Of The Review:**


See "Summary Of The Paper." Regarding the writing, methodology, and experiments, this article is far below the acceptance criteria for a good conference like ICLR.

---

> ### Author Response · Authors · 2022-11-15
> **Naive design with guassian filter to be considered only and comparison with other normalization methods**
>
> Thank you for your time and valuable comments. Idea comes shortly before the initial submission, and we try to present the preliminary results for other researchers. Our paper is about lateral inhibition inspired structure. We have obtained the code from authors of [1] (ICLR 2022), according to some reviewers' opinion, and outperform related methods by a large margin.
>
> - **The authors claim some gain in their approach. But as we all know, after introducing attention or dynamic mechanism, a general neural network can have some improvement. In particular, the worse the network, the more obvious the improvement.**
>
>    Yes, we agree. However, the main goal of our paper is to present the effectiveness of the structure which could increase the contrast and sharpness along the boundary in a flexible manner per channel from lateral inhibition perspective, with very few parameters.
>
> - **In fact, filtering is not a new thing in neural networks.**
>
>   Yes, however, using a filter from lateral inhibition perspective (consider Mach band effect) is rarely explored.
>
> - **A very working method should be validated on large-scale tasks. In addition to classification, there is detection, segmentation, and so on.**
>
>   Yes, we agree. Given limited time and resources, feature learning research may be only verified using small network on ImageNet dataset for image classification, as brain-inspired methods [1][2] (ICLR 2021, 2022). We will do the best we can.
>
> - **The writing of this article is very poor. There are multiple copies of the text in the article. For example, there are multiple repetitions of text in the abstract and introduction. A good article should have a better way of expressing it.**
>
>   Yes, given the limited time before submission. We will modify the paper and incorporate related normalization methods as well as other parts before the deadline.
>
> - **Gaussian filtering and bilateral filtering are not new in neural networks. The method proposed in the article is equivalent to a bilateral filtering non-local neural network.**
>
>   Well, we have a learnable weight W for our guassian filter, which is flexible (could be zero, positive or negative) to fit various lateral interactions). *The convolution result will be subtracted from the original feature map.*
>
> - **The article associates its method with depth-wise convolution. This is very imprecise and arrogant. If this is true, is MLP-Mixer also a special case of this article?**
>
>   Please ignore the depth-wise design. Our naive guassian design could increase the contrast and sharpness along the boundary in a flexible manner, while making homogeneous area less prominent (consider Mach band effect). *The convolution result from lateral inhibition perspective will be subtracted from the original feature map.*
>
> - **As I said before, adding attention or dynamics to the neural network, such as senet, sknet, acnet, dynamic nets, will bring gains. There are too many such examples. So the results in Table 2 are not dazzling at all.**
>
>   Well, design from lateral inhibition perspective (consider Mach band effect) for spatial adjustment to increase the contrast and sharpness along the boundary (in a flexible manner) is rarely explored. It is more brain-similar and may provide some value for researchers.
>
> - **Comparison.**
>
>   We have requested and obtained the code from [2] (and fix their missing padding for the initial convolution) for comparison with full size AlexNet (initially our method uses Pytorch build-in official model for AlexNet, which comes from [3] with less channels). AlexNet is the only model used in [1][2] (ICLR 2021, 2022) on ImageNet dataset. Normalization usually takes the form with a scaling factor, shift mean, and bias. With a simple scaling factor (to be learned) per channel, our method already outperforms all comparison methods; with shift mean or bias (to be learned) per channel makes further improvement. Results on ImageNet dataset (best model for 90 epochs as [1], while 100 epochs show no much difference) for Top-1 Acc with 4 GPU setting:
>
>     - AlexNet (baseline) 52.82%
>     - GroupNorm    57.76%
>     - LayerNorm    58.81%
>     - BatchNorm    59.04%
>     - WeightedNorm-c (center)  59.41%
>     - DivisiveNorm  59.54%
>     - Ours (scale) 59.87%
>     - Ours (shift-scale) 60.26%
>     - Ours (scale-bias) **60.39%**
>
> [1] Miller, M., Chung, S., & Miller, K. D. (2021, September). Divisive Feature Normalization Improves Image Recognition Performance in AlexNet. In International Conference on Learning Representations.
>
> [2] Pan, X., Giraldo, L. G. S., Kartal, E., & Schwartz, O. (2021). Brain-inspired weighted normalization for CNN image classification. In International Conference on Learning Representations.
>
> [3] One weird trick for parallelizing convolutional neural networks. Alex Krizhevsky, 2014.

---

> ### Comment · Reviewer_GVUz · 2022-12-11
> **Responses to the authors' responses:**
>
>
> I am very grateful to the authors for their thoughtful responses to all reviewers. This is very commendable. I really admire that.
>
> As I said in the comments, the exploratory spirit of this paper, specifically the exploratory spirit of exploring neural networks that conform to human cognition, is worthy of recognition. I think this is a very important direction. I look forward to and believe that the authors can make outstanding contributions in this direction.
>
> Although the authors may have rushed to submit this paper, the spirit of the authors is worthy of recognition. I look forward to the authors doing impactful work.

---

### Official Review · Reviewer_edpQ · 2022-10-24

**Confidence:** 5
**Clarity, Quality, Novelty And Reproducibility:** 1\ Why the low-pass filter with elimi…
**Correctness:** 3
**Technical Novelty And Significance:** 3
**Empirical Novelty And Significance:** 2
**Recommendation:** 6

**Strength And Weaknesses:**

The lateral inhibition based idea to explore the usability of the lateral inhibition into artificial neural network is interesting and useful. Such an approach should be strongly supported.

The key weakness of this paper is that there are no any explanations why their design improves the performance.

**Summary Of The Paper:**

This paper incorporates the lateral inhibition of neuron into the Deep network for image classification. Results show the effectiveness of the proposed mechanism. The lateral inhibition is not novel but the idea to explore the usability of the lateral inhibition into artificial neural network is interesting.  The key weakness of this paper is that there are no any explanations why their design improves the performance.


**Summary Of The Review:**

It can be seen that the work is not complete with only five pages as the main text. Many important issues such as explaining why the design of lateral inhibition improves the performance. Computationally, what’s the difference between channel attention and the inhibition computation?  In summary, the technique contribution is simple.

I encourage authors to enhance their work by addressing 1-2 key questions mentioned above, adding content and depth to the paper.

---

> ### Author Response · Authors · 2022-11-15
> **Why the low-pass filter with eliminating the central weight could model the lateral inhibition**
>
> Thank you for your time and valuable comments, as well as related models [1][2][3]. We will address your concerns accordingly.
>
> - **The key weakness of this paper is that there are no any explanations why their design improves the performance.**
>
>   It could increase the contrast and sharpness along the boundary in the same channel (consider Mach band effect). On one side near the boundary, strong  neighbors become weak to have reduced inhibition, the remaining value gets sudden increase. While on the other side near the boundary, remaining value gets sudden decrease.  Also we have a learnable weight W for our guassian filter, which is flexible (could be zero, positive or negative) to fit various lateral interactions.
>
> - **Why the low-pass filter with eliminating the central weight could model the LATERAL INHIBITION.**
>
>   We model lateral inhibition from neurobiology findings, that inhibition strength decreases with increased distance. Gussian (low-pass) filter could reflects the decay regarding the distance and is widely used (e.g., smoothing in image processing, multi-scale representation in Scale-space theory), as well as for lateral inhibition influence in space (the forms often assumed in theoretical calculations in [5].
>
>   Our guassian inhibition design follows common lateral inhibition description that an excited neuron can reduce the activity of its neighbors, by taking inhibition from neighbors out of the original value. It can be viewed a special difference of Gaussian (DoG), with two gaussian filters of size 1x1 and 3x3, but central weight eliminated for 3x3 filter. For the case of one activate neuro with no activate neighbors, it receives zero inhibition from neighbors and remains the same value in our design, but decreases a lot with central weight.
>
>
> - **Eq (5) seems an OFF-receptive field?**
>
>   The convolution result needs to be subtracted from the original feature map in our method, so finally it is On-receptive field. For comparison purpose, we only focus on our guassian filter (naive design) with weight W. When W is positive, finally it is On-receptive field.
>
>
> - **The inhibition computation by subtracting a computed value from the previous one is commonly used in many previous models [1][2][3]. The traditional models simulated visual mechanisms should be clearly mentioned and discussed. The description of related work is very limited as the reason shown in above.**
>
>   Yes, they will be mentioned and discussed. Our method could increase the contrast and sharpness along the boundary in a flexible manner (while making homogeneous area less prominent for suppression), then the following convolution in the original AlexNet could combine different channels to learn different antagonistic response as in [1][3].
>
>   For [2] (ICLR 2021), it does normalization along the channel dimension for their WNc (WeightedNorm-center) method. Our spatial lateral inhibition method outperform it. They also add spatial normalization after channel normalization, leading to decreased performance on ImageNet as stated in [2].
>
> - **The learned weight W plays the similar function of channel attention. Hence, what’s the difference between channel attention and the inhibition computation?**
>
>   The learned weight W decides the amplitude of our guassian filter and final value after spatial adjustment (before subtraction from original value). Channel attention aims to scale one entire channel with one value (by multiplication), which is learned using MLP on the average value of all channels. Channel attention has no spatial adjustment within that channel, which we do for each location individually (consider Mach band effect).
>
> - **Eq (5) seems an OFF-receptive field?  Author should visualize the learned kernels as many examples.**
>
>   We focus on our naive design of guassian filter (central weight eliminated), with learnable weight W for comparison.
>
> - **Comparison.**
>
>   Normalization usually takes the form with a scaling factor, shift mean, and bias. For comparison, with a simple scaling factor (to be learned) per channel, our method already outperforms all comparison methods; with shift mean or bias (to be learned) per channel makes further improvement. Results on ImageNet dataset (best model for 90 epochs as [2], while 100 epochs show no much difference) for Top-1 Acc with 4 GPU setting:
>
>     - AlexNet (baseline) 52.82%
>     - GroupNorm    57.76%
>     - LayerNorm    58.81%
>     - BatchNorm    59.04%
>     - WeightedNorm-c  59.41%
>     - DivisiveNorm  59.54%
>     - Ours (scale) 59.87%
>     - Ours (shift-scale) 60.26%
>     - Ours (scale-bias) **60.39%**
>
> [4] Miller, M., Chung, S., & Miller, K. D. (2021, September). Divisive Feature Normalization Improves Image Recognition Performance in AlexNet. In International Conference on Learning Representations.
>
> [5] On Tuning and Amplification by Lateral Inhibition. Proceedings of the National Academy of Sciences of the United States of America,
> Vol. 62, No. 3 (Mar. 15, 1969), pp. 733-740.

---

### Official Review · Reviewer_Ct44 · 2022-10-25

**Confidence:** 5
**Correctness:** 2
**Technical Novelty And Significance:** 1
**Empirical Novelty And Significance:** 1
**Recommendation:** 3

**Clarity, Quality, Novelty And Reproducibility:**

The proposed work is not novel and evaluations are preliminary. The authors don't discuss sharing code or trained models, which is troublesome to reproduce the presented results.

**Strength And Weaknesses:**

Strengths:
1) There are considerable gains on AlexNet and ResNet while using LI, however, I would like to note that these are preliminary results as also highlighted by the authors.

Weaknesses:
1) The proposed work lacks novelty, several methods in the past have tried to apply a very similar lateral or divisive inhibition mechanism to deep convolutional networks and have reported gains in image classification performance (particularly when added to AlexNet). See [1] [2] and [3] for example. The proposed work is almost exactly similar to [1].
2) The evaluation is very preliminary and lacks comparison to suitable baselines (other kinds of normalization such as BatchNorm, LayerNorm, etc.) or other normalization techniques such as [1, 2] which are very relevant.

References:
1. Hasani, H., Soleymani, M., & Aghajan, H. (2019). Surround modulation: A bio-inspired connectivity structure for convolutional neural networks. Advances in neural information processing systems, 32.
2. Miller, M., Chung, S., & Miller, K. D. (2021, September). Divisive Feature Normalization Improves Image Recognition Performance in AlexNet. In International Conference on Learning Representations.
3. Pan, X., Giraldo, L. G. S., Kartal, E., & Schwartz, O. (2021). Brain-inspired weighted normalization for CNN image classification. bioRxiv.


**Summary Of The Paper:**

The authors propose to add a biologically inspired lateral inhibition mechanism into deep convolutional networks for image recognition. When incorporated into AlexNets and ResNets, LI seems to improve performance on ImageNet classification without increasing trainable parameters. The authors examine the LI filter weights and find a biologically-resemblant center-surround pattern of inhibition.

**Summary Of The Review:**

The proposed work is preliminary and lacks the novelty and quality of work expected at ICLR. I do not recommend accepting this paper at this stage. I suggest the authors to please consider a suitable workshop for this work and a significant extension of this work could be suitable for the ICLR audience with wider evaluation using suitable baselines and tasks.

---

> ### Author Response · Authors · 2022-11-13
> **Novelty and comparison with other methods**
>
> Thank you for your time and valuable comments, especially recent brain-inspired methods [3] (ICLR 2021), [2] (ICLR 2022). Idea comes shortly before the initial submission, and we try to present the preliminary results for other researchers. We will address your concerns accordingly. Paper will be revised to highlight our contribution and incorporate the related methods as well as comparison results.
>
> - **Novelty.**
>
>     (i) Most comparison methods in [2], [3] are normalization across different 2D channels at the same 2D spatial location (x, y), without spatial adjustment exploring neighbors within the same channel, which our method does (could increase the contrast and sharpness along the boundary in a flexible manner, while making homogeneous area less prominent, consider Mach band effect) to be more brain-similar. The only spatial normalization part (called surround norm) in WeightedNorm [3] is applied after channel normalization (called WeightedNorm-c, center norm). This combined design has the name of WeightedNorm-cs (center surround norm), and it has even decreased performance (than without spatial normalization) on ImageNet dataset according to [3].
>
>
>     (ii) Things need to be noticed from [1] (SM, surround modulation) (NIPS 2019) and our method:
>
>       a) [1] applies two gaussian filters (5x5) with 1:1 weight (fixed) for the difference of Gaussian (DoG) function. While we use apply one gaussian filter (3x3, central value as 0), with the convolution result subtracted from the original feature map (could be viewed as two filters with size 1x1 and 3x3), to replace the two-gaussian-filter design. Our method has 1:w weight, where w is learnable (flexible) per channel for the amplitude of our (3x3) gaussian filter (can be zero, positive or negative) to fit various lateral interactions.
>
>       b) [1] only applies SM filter on half of the initial channels. Applying it on full channels leads to reduced performance, which indicates its limitation of flexibility.
>
> - **Comparison.**
>
>   We have requested and obtained the code from [2] (and fix their missing padding for the initial convolution) for comparison with full size AlexNet (initially our method uses Pytorch build-in official model for AlexNet, which comes from [4] with less channels). Normalization usually takes the form with a scaling factor, shift mean, and bias. For comparison, with a simple scaling factor (to be learned) per channel, our method already outperforms all comparison methods; with shift mean or bias (to be learned) per channel makes further improvement. Results on ImageNet dataset (best model for 90 epochs as [2], while 100 epochs show no much difference) for Top-1 Acc with 4 GPU setting:
>
>     - AlexNet (baseline) 52.82%
>     - GroupNorm    57.76%
>     - LayerNorm    58.81%
>     - BatchNorm    59.04%
>     - WeightedNorm-c  59.41%
>     - DivisiveNorm  59.54%
>     - Ours (scale) 59.87%
>     - Ours (shift-scale) 60.26%
>     - Ours (scale-bias) **60.39%**
>
> [4] One weird trick for parallelizing convolutional neural networks. Alex Krizhevsky, 2014.

---

> > ### Comment · Reviewer_Ct44 · 2022-11-25
> > **Acknowleding author responses**
> >
> > Dear Authors,
> >
> > I thank you very much for carefully considering comments in my review and adding meaningful baselines to the comparisons. I appreciate you highlighting the differences between the proposed work and Hasani et al [2019]. However, I don't see this comparison being made in the new set of results in your response above. I thank you for adding DivisiveNorm and WeightedNorm comparisons to the response. However, I still think the results are quite preliminary and aren't sufficient in my opinion to be accepted at ICLR at this moment.
> >
> > I hope the authors please work on the following two broad directions to add more value to their submission: (1) Understanding Lateral Inhibition / its relevance to biology -- I see that authors show  improved performance on ImageNet with the AlexNet architecture. Why does LI cause this improved performance? What is different about the features learned by AlexNet with LI vs without LI? I find the explanation related to Mach Band effect interesting but the authors need to demonstrate if it is indeed what is happening in the network; (2) Relevance to deep learning -- If the authors intend to make this submission relevant to deep learning audience, they must significantly strengthen empirical results and show across multiple networks and (datasets or tasks) that the proposed LI module produces gains with error bars across multiple random initializations.
> >
> > At this stage, I retain my score and do not recommend acceptance. If the authors were to work on either or both of the above directions, I'm sure this valuable contribution will be further strengthened and will eventually be of interest to machine and/or biological vision audience. Once again, thank you very much to the authors for engaging in the rebuttal phase and good luck on enhancing this submission.

---

> > > ### Author Response · Authors · 2022-11-27
> > > **For recent review**
> > >
> > > Thank you for your valuable time and review. Just a little bit more information for your concerns:
> > > - **I appreciate you highlighting the differences between the proposed work and Hasani et al [2019]. However, I don't see this comparison being made in the new set of results in your response above.**
> > >
> > >     For Hasani et al [2019] on ImageNet dataset with AlexNet (the original paper only using a subset of 100 categories in reduced resolution of 160x160, with 500 instances for training and 100 instances for testing per category, on a small convolutional network), when applying the fixed difference of Gaussian (DoG) for the all the initial convolutional channels, the Top-1 accuracy is 51.46%, when applying to all channels of all convolutional layers, the Top-1 accuracy is 51.39%. Results indicate reduced accuracy by adding their filter to AlexNet on the original size and resolution of ImageNet dataset, which could be caused by lacking flexibility of the filter (as the original paper says applying to half of the initial channels is better than applying to all channels, but both improve the accuracy). And applying to all channels at all convolutional layers further decreases the accuracy. While our method applying to all layers significantly improves the accuracy to 60.39%.
> > >
> > > - **Why does LI cause this improved performance?**
> > >
> > >     The main idea of our lateral inhibition inspired design is to calculate lateral inhibition value from neighbors at each location for each channel, then subtracts that value from the original value at each location. With learnable weight W, it is flexible (could be zero, positive or negative) to model different lateral interaction (zero for none, positive for lateral inhibition, negative for lateral enhancement).
> > >
> > >     Our guassian inhibition inspired design follows common lateral inhibition description that an excited neuron can reduce the activity of its neighbors, by subtracting inhibition caused by neighbors from the original value. It can be viewed a special difference of Gaussian (DoG), with two gaussian filters of size 1x1 and 3x3, but central weight eliminated for 3x3 filter. For the case of one activate neuro with no activate neighbors, it receives zero inhibition from neighbors and remains the same value in our design, but decreases a lot with central weight.
> > >
> > >     Our LI design **could increase the contrast and sharpness along the boundary** in the same channel (consider Mach band effect). **When calculating, on one side near the boundary, strong neighbors become weak to have reduced inhibition, the remaining value gets sudden increase. While on the other side near the boundary, remaining value gets sudden decrease. This could make larger contrast than without it** (see the the illustration of Mach bands in our paper, with sudden curves towards two different directions along both sides of the boundary, to make the boundary more prominent).
> > >
> > >     **Our learnable weight W is flexible to (can be zero, positive or negative) to fit various lateral interactions.** For our method, by manually checking the learnable W at all layers (in total 5 layers with only one convolution per layer for AlexNet), it shows that most of the channels have positive weight for that channel from layer 1 ~ layer 3, meaning lateral inhibition happens (original value minus the multiplication value of positive weight and positive RELU activation value from neighbors) for these early layers in this network, which could increase the contrast and sharpness along the boundary. Then for layer 4 ~ layer 5, most of the channels have negative weight for that channel, meaning neighbors could enhance the activation (in biological findings, there are lateral interactions which neighbors inhibit or strengthen the signal). It makes sense that for early layers the network needs more spatial-accurate signals, but for high level layers, the network needs the signals (larger parts, concept) in large receptive field to combine all of them for recognition.

---

### Decision · Program_Chairs · 2023-01-20

**Decision:**

Reject

**Justification For Why Not Higher Score:**

There is consensus in rejecting this work.

**Justification For Why Not Lower Score:**

N/A

**Metareview: Summary, Strengths And Weaknesses:**

The paper proposes to add a lateral inhibition mechanism into CNN and shows performance improvements when paired with AlexNets and ResNets. There is consensus among the reviewers on the novelty of the approach and with evaluation that is still preliminary and not as thorough as one would expect from an application work I am afraid I have to reject it.
I encourage the authors to continue with their study and to provide more compelling empirical to potentially strengthen their contribution.